# Inactivation of Bacteria and Residual Antimicrobials in Hospital Wastewater by Ozone Treatment

**DOI:** 10.3390/antibiotics11070862

**Published:** 2022-06-27

**Authors:** Takashi Azuma, Miwa Katagiri, Tsuyoshi Sekizuka, Makoto Kuroda, Manabu Watanabe

**Affiliations:** 1Department of Pharmaceutical Sciences, Osaka Medical and Pharmaceutical University, Takatsuki 569-1094, Japan; takashi.azuma@ompu.ac.jp; 2Department of Surgery, Toho University Ohashi Medical Center, Tokyo 153-8515, Meguro-ku, Japan; girioyoshikazu0513@gmail.com; 3Pathogen Genomics Center, National Institute of Infectious Diseases, Tokyo 162-8640, Shinjyuku-ku, Japan; sekizuka@niid.go.jp

**Keywords:** hospital wastewater, ozone treatment, metagenomics, *enterobacteriaceae*, extended-spectrum *β*-lactamase (ESBL), carbapenemase, residual antimicrobials

## Abstract

The emergence and spread of antimicrobial resistance (AMR) has become a persistent problem globally. In this study, an ozone treatment facility was established for an advanced hospital wastewater treatment in a core hospital facility in an urban area in Japan to evaluate the inactivation of antimicrobial-resistant bacteria and antimicrobials. Metagenomic DNA-seq analysis and the isolation of potential extended-spectrum *β*-lactamase (ESBL)-producing bacteria suggested that ozone exposure for at least 20 min is required for the adequate inactivation of DNA and ESBL-producing bacteria. *Escherichia coli* and *Klebsiella* species were markedly susceptible to 20-min ozone exposure, whereas *Raoultella ornithinolytica* and *Pseudomonas putida* were isolated even after an 80-min exposure. These ozone-resistant bacteria might play a pivotal role as AMR reservoirs in the environment. Nine antimicrobials (ampicillin, cefdinir, cefpodoxime, ciprofloxacin, levofloxacin, clarithromycin, chlortetracycline, minocycline, and vancomycin) were detected at 373 ng/L to 27 μg/L in the hospital wastewater, and these were removed (96–100% removal) after a 40-min treatment. These results facilitate a comprehensive understanding of the AMR risk posed by hospital wastewater and provides insights for devising strategies to eliminate or mitigate the burden of antimicrobial-resistant bacteria and the flow of antimicrobials into the environment. To the best of our knowledge, this is the first report on the implementation of a batch-type, plant-scale ozone treatment system in a hospital facility to execute and evaluate the inactivation of drug-resistant bacteria and antimicrobials.

## 1. Introduction

Research on health and environmental risk assessment and countermeasures against antimicrobial resistance (AMR) is ongoing on a global scale. The spread of AMR is a cause of increasing concern for the future use of antimicrobials and other measures to control infectious diseases. AMR is an important issue requiring immediate and effective action by the World Health Organization (WHO) and the G7 group of industrialized nations. In recent years, it has been observed that as soon as a new antimicrobial is used in clinical practice, resistant bacterial strains emerge, which has resulted in the spread of multiple antimicrobial-resistant strains. In addition to the clinical use of antimicrobials, community-acquired infections caused by healthy carriers and outbreaks of AMR, originating from livestock, fisheries, and other industries, are also becoming problematic, making it essential to comprehensively understand the trend of AMR on a global scale and a “One Health” approach is desired to combat these problems [1,2,3].

Antimicrobial-resistant bacteria and antimicrobial residues have been reported in medical effluents from hospitals and other medical facilities [4,5]. Conventional wastewater treatment, which aims to remove organic matter derived from human waste, as reflected by biochemical oxygen demand, could not completely remove environmental pollutants. The discharge of treated water into aquatic environments, such as rivers, lakes, and oceans, creates new environmental pollution problems [5,6]. The water pollution control law in Japan for discharges to general wastewater by organizations, such as hospitals, regulates pH, heavy metals, the coliform group, etc., but does not set regulatory values for pathogenic bacteria or antimicrobial-resistant bacteria, as in other countries [7,8]. At municipal wastewater treatment plants located in Japan, treated wastewater is discharged in compliance with the regulations on the coliform group (<3000 viable bacteria count/mL). However, in areas where a confluence system is adopted, rainwater flows into the same waterway as sewage and the wastewater is discharged into rivers and the ocean without sufficient treatment, during rainy weather. It is, therefore, important to understand the actual situation of AMR originating from hospital wastewater, evaluate its risk, and consider effective countermeasures for assessing and solving the AMR problem, not only at the environmental site but also at the clinical site [7,8,9,10,11].

In the abovementioned context, the treatment of AMR-related factors in hospital wastewater before discharging it into the sewage system could be one of the measures for the effective reduction in AMR. In addition, it is expected that information on AMR inherent in hospital wastewater, as environmental AMR in the medical field, will not only support nosocomial infection control measures for the early detection and prevention of nosocomial infection strains that may occur, but will also enable discussions on the fundamental measures to be taken against AMR. However, owing to the associated difficulties, limited studies have been conducted on hospital wastewaters worldwide [7,8,12].

Various wastewater treatment systems that could be effective in treating hospital wastewater have been developed and studied; these include systems involving the use of the Fenton process [13,14], electrolysis [15], TiO_2_ [16], persulfate [17], UV/chlorine [18], and ozone [14,19]. Ozone treatment has been the focus of much research in recent years because it does not require the addition of any chemicals, the wastewater is free from residues after treatment, and the treatment has strong sterilizing and deodorizing effects [20]. However, the efficacy of ozone treatment has been primarily evaluated in small-scale test systems in laboratories and has yet to be studied on the actual hospital wastewater scale. As such, there is a lack of knowledge on the practical application of ozone treatment [21,22,23]. If the ozone treatment of hospital wastewater on a practical scale is demonstrated to be a reasonable solution for the environmental AMR concern, it will be possible to solve the problem of AMR and contribute to the One Health approach [24]. Furthermore, the results can potentially contribute to the public interest in regional security and would guarantee the safety of the local population [25,26]. Toward this end, in the present study, a plant-scale ozone treatment system was implemented in a hospital facility using a batch-type process to execute and evaluate the inactivation of antimicrobial-resistant bacteria and antimicrobial residues and mitigate the environmental impact of AMR.

## 2. Materials and Methods

### 2.1. Sample Collection

Hospital wastewater samples were collected at the Ohashi Medical Center (BN; 35.652578°N, 139.683959°E), with a capacity of 319 beds, in Toho University, located in Jonan area, Tokyo, Japan. In the hospital, various wastewater types (stool and urine), generated as a result of hospital activities, are stored in two underground wastewater tanks, with an effective volume of 22.5 m^3^, without mixing with other drainage. The supernatant is pumped directly into the public sewage system several times a day, and the settled sediments are collected and incinerated by a specialized waste management company. It was impossible to quantify the daily inflow and outflow of wastewater tanks because of the lack of any system for regular measurements.

### 2.2. Ozone Treatment

Inactivation of bacteria and antimicrobials present in the hospital wastewater by ozonation was performed using an ozone treatment system installed in the hospital facility. Wastewater from one of the two storage tanks was introduced into a wastewater treatment tank, with an effective volume of 1 m^3^, for semi-batch ozone treatment. The appearance and configuration of the ozone treatment system used in this study are shown in Figure 1.

Ozone was generated using an ozone generator (Ozonia^®^ TOGC45X, Suez Environment, Paris, France). The hospital wastewater in the wastewater treatment tank was circulated using a circulation pump (32LPS5.75E, Ebara Corporation, Tokyo, Japan) at a flow rate of 80 L/min, and ozone gas was fine-bubbled through fine-bubble generating nozzles (EE091, For EARTH Co., Ltd., Tokyo, Japan) and introduced into the wastewater treatment tank. The ozone treatment was performed at an ozone generation rate of 34 g/h and an effective ozone gas concentration of 111 mg/L. The experiments were initiated by sparging ozone gas continuously into the filled wastewater treatment tank. A portion (100 mL) of the solution in this tank was sampled at 0, 10, 20, 40, and 80 min after the start of the experiment. These durations were determined based on the average contact times in Japanese wastewater treatment plants that implement ozonation before discharging their effluent into rivers [27] and by considering previously reported values [11,28]. Dissolved ozone and gaseous ozone concentrations and water quality parameters (COD, BOD, SS, and the coliform group) measured during ozonation prior to conducting this study is shown in Appendix A. Sodium thiosulfate or tryptic soy broth was immediately added to mitigate the effects of any residual ozone in the samples [29,30]. The samples were then stored at 4 °C in the dark and further processed within 12 h.

### 2.3. Metagenomic DNA-Seq Analysis of Wastewater Samples

To collect organisms larger than bacteria, ozone-treated wastewater samples were passed through TPP Rapid Filtermax Vacuum Filtration systems (TPP, Trasadingen, Switzerland) in 500 mL bottles fitted with 49 cm^2^, 0.2 µm polyethersulfone membranes. The membranes were removed from the bottles and stored at −30 °C until DNA extraction. One-fourth of the collected membrane was cut into small pieces and placed in ZR-96 BashingBead Lysis Tubes (0.1 and 0.5 mm; Zymo Inc., Irvine, CA, USA). Bacterial lysis buffer (800 µL; Roche, Basel, Switzerland) was added to the bead tube, which was frozen at −30 °C and thawed at 23 °C. The tube was subjected to bead-beating (1500 rpm for 10 min) using a GenoGrinder 2010 homogenizer. After brief centrifugation (8000× *g* for 3 min), 400 µL of the supernatant was collected. The DNA in the supernatant was purified using a Roche MagNa Pure Compact instrument (DNA_Bacteria_v3 protocol; Elution: 50 µL). DNA concentrations and purity were measured using the Qubit DNA HS kit (Thermo Fisher Scientific, Waltham, Massachusetts, USA).

Metagenomic DNA-Seq libraries were prepared using the QIAseq FX DNA library kit (Qiagen, Hilden, Germany), followed by short-read sequencing using the iSeq platform (2 × 150-mer paired-end) (Illumina, San Diego, CA, USA). Adapter and low-quality sequences were trimmed using Sickle version 1.33 (https://github.com/najoshi/sickle), considering the following parameters: average quality threshold “-q 20” and minimum length threshold “-l 40” Metagenomic DNA-Seq analysis was performed using cleaned reads for homology search without de novo assembly in all subsequent analyses. Detailed scripts and databases are described below.

Taxonomic classification of every single read from metagenomic analysis was performed using mega-BLAST (e-value threshold, 1E^−20^; identity threshold, 95%) against the NCBI nt database using MePIC2 [31], and subsequently analyzed using MEGAN 6 [32].

### 2.4. Resistome Analysis

Metagenomic DNA-seq analysis was performed using cleaned reads for homology searches without de novo assembly. Before resistome analysis, an ARG database was constructed using the bacterial antimicrobial resistance reference gene (National Center for Biotechnology Information (NCBI) BioProject ID, PRJNA313047) and ResFinder (https://bitbucket.org/genomicepidemiology/resfinder_db/src/master/). The study database was constructed using Makeblastdb in the basic local alignment search tool (BLAST+). The operational taxonomic units (OTUs) in the ARG database (AMROTU ver. 2022-04-11) were created by clustering at ≥90% sequence identity and ≥80% coverage using vsearch version 2.10.4. The metagenomic DNA-seq reads were searched using mega-BLAST (e-value threshold, 1E^−20^; identity threshold, 95%) against the customized ARG database. The detected genes were summarized for each OTU of the ARGs. Reads per kilobase of gene per million (RPKM) counts were calculated using the following formula for normalization:

RPKM = number of detected reads against OTUs/[average gene length of detected OTUs (bp) × total number of trimmed reads] × 10^9^.

### 2.5. Whole-Genome Analysis of Bacterial Isolates

Whole-genome sequencing of bacterial isolates was performed using the NextSeq 1000 platform (Illumina). The draft genome sequence was assembled using A5-miseq with Illumina short-read data. Gene annotation was performed using DFAST version 1.2.3 [33] using the following databases: DFAST default database, ResFinder database [34], Bacterial Antimicrobial Resistance Reference Gene (BARRG) database (PRJNA313047), and Virulence Factors Database [35]. Multilocus sequence typing (MLST) was performed using “mlst” program version 2.16.2 (Seemann T, mlst Github https://github.com/tseemann/mlst) with PubMLST database (https://pubmlst.org/).

### 2.6. Analytical Procedures for Antimicrobials

A total of 15 antimicrobials were investigated on the basis of a previous report on their concentrations and detection frequencies in hospital effluent, wastewater, and river water, both in Japan and around the world, as well as on the basis of antimicrobial use in Japan [6,36,37,38]. *β*-lactams (ampicillin, cefdinir, cefpodoxime, cefpodoxime proxetil, and ceftiofur), new quinolones (ciprofloxacin and levofloxacin), macrolides (azithromycin and clarithromycin), tetracyclines (chlortetracycline, doxycycline, minocycline, oxytetracycline, and tetracycline), and glycopeptide (vancomycin) (>98%) were targeted in the present study.

The concentrations of target antimicrobials in the wastewater were determined using a combination of solid phase extraction (SPE) and ultra-performance liquid chromatography–tandem mass spectrometry, as described previously [36]. Briefly, 10 mL of wastewater was filtered through a glass-fiber filter (GF/B, 1 μm pore size, Whatman, Maidstone, UK). The solutions were then passed through SPE cartridges (OASIS HLB, 200 mg; Waters Corp., Milford, MA, USA) at a flow rate of 1 mL/min. The cartridges were washed with 6 mL of Milli-Q water, preadjusted to pH 3, and then dried using a vacuum pump. Finally, the adsorbed antimicrobials were eluted with 3 mL acetone and 3 mL methanol or with 2 mL of 10% (*v*/*v*) formic acid in acetone, 2 mL of 10% (*v*/*v*) formic acid in methanol, and 2 mL of 5% ammonia–methanol (*v*/*v*). Each combined eluted solution was evaporated mildly to dryness under a gentle stream of N_2_ gas at 37 °C. The residue was solubilized in 200 μL of a 90:10 (*v*/*v*) mixture of 0.1% formic acid solution in methanol, and 10 μL of this solution was subjected to analysis using a UPLC system coupled to a tandem quadrupole mass spectrometer (TQD, Waters Corp.), equipped with an electrospray ionization source operated in positive ion mode.

Quantification was performed by subtracting the blank data from the corresponding data yielded by the spiked sample solutions to account for matrix effects and losses during sample extraction [39,40]. The recovery rates of antimicrobials in the wastewater influent ranged from 48% to 98% (Appendix A) The limits of detection (LODs) and limits of quantification (LOQs) were calculated as the concentrations at signal-to-noise ratios of 3 and 10, respectively [41,42]. These values are also summarized in Appendix A.

## 3. Results

### 3.1. Proportion of Bacteria in Hospital Wastewater after Ozone Treatment

Hospital wastewater (1 m^3^) in a batch reactor was treated with ozone fine bubbles (Figure 1), and the treated sample was subjected to metagenomic DNA-seq analysis (Table 1). Exposure to ozone for 20 min was sufficient to inactivate bacteria to <0.01% of the original number (for instance, those of the genus *Bacteroides*). Other bacteria genera were also reduced up to 0.02% of the original number upon 20-min exposure, indicating that most organisms were effectively inactivated in a short duration of exposure (Table 1). Besides bacterial genera, we also investigated the value of RPKM for each operational taxonomic unit (OTU) of an AMR gene (ARG) after ozone treatment (Appendix A). In original raw wastewater before ozone treatment, the *tet*(Q) gene that is related to *Bacteroides* species in human feces was predominantly detected and the *bla*_GES-1_ variants that are related to environmental bacteria such as *Aeromonas* and *Pseudomonas* species were secondly predominant. After 20 min of exposure to ozone, all ARGs were not detected in metagenomic DNA-Seq analysis (Appendix A).

Gram-negative *Enterobacteriaceae* from urine and feces were cultured on B.T.B. agar (Bromothymol Blue, lactose agar; Drigalski Agar, Modified) and colony forming units (CFU) were determined. The percentage of viable bacteria was reduced to 76.2% (179,000/235,000 CFU) and 11.9% (27,900/235,000 CFU) upon a 10- and 20-min exposure to ozone, respectively (Figure 2).

To determine the efficacy of the ozone microbubble treatment in inactivating potential *β*-lactam-resistant bacteria, the treated sample was spread on a CHROMagar ESBL plate, and the number of pigmented colonies was counted (Figure 2). Based on the CFU values, the percentage of all bacteria was reduced to 56.7% (51,000/90,000 CFU) and 20.2% (18,200/90,000 CFU) upon a 10- and 20-min exposure, respectively (Figure 2). The percentage of colonies showing dark blue or pink pigmentation was reduced to 5.6% (3100/55,000 CFU) and 0.8% (100/12,000 CFU), respectively, upon a 20-min exposure, whereas that of white colonies (no pigmentation) was not significantly reduced (65.2% upon a 20-min exposure) compared with that of the abovementioned pigmented colony types.

### 3.2. Susceptibility of Bacterial Species in Hospital Wastewater to Ozone Treatment

To elucidate the susceptibility of different bacterial species to ozone treatment, a markedly pigmented colony from each time-point was selected for whole genome sequence analysis (Appendix A). Based on the results described in Section 2.1, the 20-min treatment was considered a reasonable time-point to investigate the susceptibility of notable bacterial species to ozone. CTX-M-producing *E. coli* isolates (pink pigmentation on CHROMagar ESBL plate) were found to be highly susceptible to ozone treatment compared with other isolates (Figure 2 and Appendix A). Although variable ESBL/carbapenemase-producing *Enterobacteriaceae* isolates were identified up to 20 min after treatment, *Raoultella ornithinolytica* (dark blue pigmentation) and *Pseudomonas putida* (white, no pigmentation) were remarkably isolated even at the 40- and 80-min time points of ozone treatment (Appendix A).

### 3.3. Removal of Antimicrobials by Ozone Treatment

Nine antimicrobials (ampicillin, cefdinir, cefpodoxime, ciprofloxacin, levofloxacin, clarithromycin, chlortetracycline, minocycline, and vancomycin) were detected at a wide range of concentrations (from ng/L to μg/L levels; 373 ng/L to 27 μg/L) in the hospital wastewater (Table 2).

The higher concentrations of antimicrobials were consistent with those previously reported from other countries [10,43,44]. The removal of antimicrobials detected in hospital wastewater by ozone treatment is summarized in Figure 3. The results show that ozone treatment reduced the residual antimicrobials in the wastewater over time, with a 96–100% removal of all targeted antimicrobials at 40 min after the treatment. The removal rate of cefdinir, levofloxacin, chlortetracycline, and vancomycin reached 90% within 10 min after the treatment, suggesting that these components are rapidly removed by ozone treatment. The removal rates of cefpodoxime, ciprofloxacin, and minocycline were 69% (cefpodoxime), 73% (ciprofloxacin), and 54% (minocycline) at 10 min after the start of treatment, and the removal rate of all components reached over 99% at 20 min after the treatment. On the contrary, ampicillin and clarithromycin were detected (20–22%) at 20 min after the start of the treatment, but the removal rate reached 96–99% after 40 min. These results suggest that ozone treatment can effectively remove antimicrobials in hospital wastewater in a short time.

## 4. Discussion

In this study, we investigated the efficacy of ozone fine bubble treatment (Figure 1) of hospital wastewater in mitigating the environmental AMR issue and evaluated the treatment by performing metagenome DNA-seq analysis (Table 1), culture of AMR isolates (Figure 2), and quantitative measurement of residual antimicrobial agents (Table 2 and Figure 3). The results of metagenome DNA-seq analysis suggested that ozone exposure for at least 20 min is required for adequate inactivation (less than 0.02% of the original concentration) of the DNA molecule (Table 1). Complete inactivation of the DNA molecule is ideal because the residual AMR genes could be utilized by bacteria to acquire AMR through horizontal gene transfer, although these are partial DNA fragments.

In addition, viable AMR bacteria in wastewater effluents are the most important and constitute a primary cause of environmental AMR burden because these bacteria can potentially grow in the environment. Therefore, these bacteria should be controlled to limit the spread of AMR bacteria in the environment. ESBL-producing bacteria were apparently reduced by 10-min of ozone exposure in a time-dependent manner (Figure 2). Intriguingly, CTX-M-producing *E. coli* isolates (pink pigment on CHROMagar ESBL plate) were found to be significantly susceptible to ozone treatment compared to the other isolates (Figure 2 and Appendix A). Notably, *R. ornithinolytica* (dark blue pigment) and *P. putida* (white, no pigment) were isolated even at 40- and 80-min of ozone treatment (Appendix A). Both these species (or specific features of these isolates) may be intrinsically resistant to ozone or perhaps a microbial biofilm or aggregation phenotype could confer them with resistance to ozone.

*R. ornithinolytica* isolates carry *bla*_CTX-M-62_ (Appendix A), alluding to the risk of horizontal gene transfer via plasmid transconjugation. *P. putida* has a low virulence, and its isolates are negative for potential *β*-lactamases and other marked ARGs, suggesting that it is an environmental bacterium with a very low risk for pathogenesis and AMR.

These results indicate that the susceptibility to ozone varies among bacterial species. The finding that ESBL-producing pathogenic bacteria, such as *E. coli* (pink pigment) and *Klebsiella* (dark blue pigment) exhibit a marked susceptibility to ozone and is consistent with previous reports [28,45,46] supports the proposal for the installation of an ozone treatment system in hospital wastewater tanks.

The identification of ozone-resistant bacteria, such as *Raoultella* and *Pseudomonas*, in this study, implies that these bacteria might play a pivotal role as an AMR reservoir in the environment and should be extensively monitored.

Residual antimicrobials detected in hospital wastewater are thought to originate from the antimicrobials used to treat diseases in clinical settings [47]. We did not detect cefpodoxime proxetil, ceftiofur, azithromycin, oxytetracycline, and tetracycline in the wastewater samples in the present investigation probably because of the non-usage of these antimicrobials in the hospital at the time the present study was conducted or because of the fact that the concentrations of antimicrobials present in wastewater represent the concentration at a particular time of the day due to grab sampling of wastewater [48,49]. In addition, it should also be noted that some antimicrobials, such as *β*-lactam antimicrobials, are attenuated in the water environment within a few hours [50,51]. Previous studies have reported that ozone treatment generally reduces the ecotoxicity compared to that of untreated compounds [52,53,54], but toxicity may increase in some cases [54]. On the other hand, the strong oxidizing action of ozone can decrease the formation of residual intermediate products by providing a sufficient processing time and by acting in combination with catalysts such as UV and hydrogen peroxide [20,55,56]. Our results support the need for further, conclusive research performed by taking experimental, technical, regional customs, bias, and unknown factors into consideration.

Studies evaluating the removal of antimicrobial-resistant bacteria and antimicrobials from wastewater using ozone treatment have primarily focused on wastewater treatment in wastewater treatment plants [53,57,58]. In these reports, the wastewater that has undergone processes such as the removal of solids by primary treatments such as sand filtration, the removal of organic matter by secondary treatments such as biological treatments, and disinfection after discharge into the water environment has been considered [58,59,60]. Moreover, as it is generally difficult to perform studies on hospital wastewater, the diversity and prevalence of antimicrobial-resistant bacteria and antimicrobials in hospital wastewaters have been evaluated only in a few studies worldwide [7,8,12]. The present study clarifies the effect of direct ozonation on hospital wastewater without pretreatment, and the treatment included fine ozone bubbles to achieve high removal efficiency; this has not been reported previously.

The problem of environmental pollution caused by the flow and persistence of antimicrobial-resistant bacteria and antimicrobials in rivers, lakes, and sea areas via wastewater systems has been reported worldwide [4,61,62,63], with reports pointing to the possibility of toxic effects on ecosystems [47,64,65]. Recent reports have suggested the possibility of the promotion of an unexpected emergence of antimicrobial-resistant bacteria from the environment [66,67,68]. Therefore, it is important to conduct a detailed assessment of the environmental risk posed by both antimicrobial-resistant bacteria and antimicrobials and to evaluate advanced wastewater treatment techniques that are effective in reducing or eliminating these risks.

## 5. Conclusions

A plant-scale ozone wastewater treatment facility based on a batch-type process was implemented in a core hospital located in the heart of Japan, and the inactivation effect on antimicrobial-resistant bacteria and antimicrobial residues was evaluated. The ozone treatment was effective in inactivating both antimicrobial-resistant bacteria and antimicrobials in hospital wastewater. Even when direct treatment, without filtration or biological pretreatment, was employed, most of the clinically problematic antimicrobial-resistant bacteria and residual antimicrobials were inactivated within 20–40 min of direct ozone treatment. The fact that a variety of antimicrobial-resistant bacteria and antimicrobials were detected at high concentrations in the hospital wastewater is a key point related to the development and spread of AMR. The fact that these organisms and antimicrobials can be inactivated by advanced wastewater treatment is significant in terms of taking feasible and effective countermeasures for addressing AMR in the environment.

The overall results signify a novel approach for preventing the environmental risks associated with the spread of AMR and could facilitate the early detection of nosocomial infection risk and the reduction of the environmental impact of AMR. Our findings could help enhance the effectiveness of introducing advanced wastewater treatment systems, not only at wastewater treatment plants but also at medical facilities, to reduce the discharge of pollutants into rivers, thereby contributing to the safety of environmental and human health.

## Figures and Tables

**Figure 1 antibiotics-11-00862-f001:**
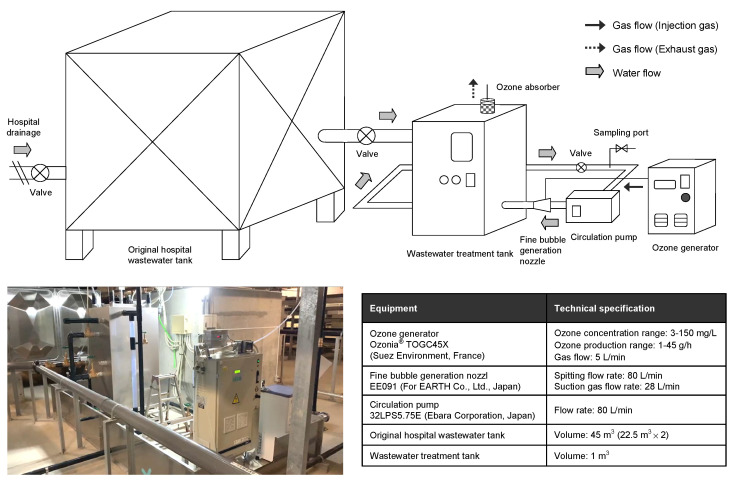
Schematic representation of the batch ozone treatment system implemented in a hospital facility. The picture shows the appearance of the advanced hospital wastewater treatment system equipped with the ozone treatment system that was tested in this study. The technical specifications of the equipment used in the system are shown in detail.

**Figure 2 antibiotics-11-00862-f002:**
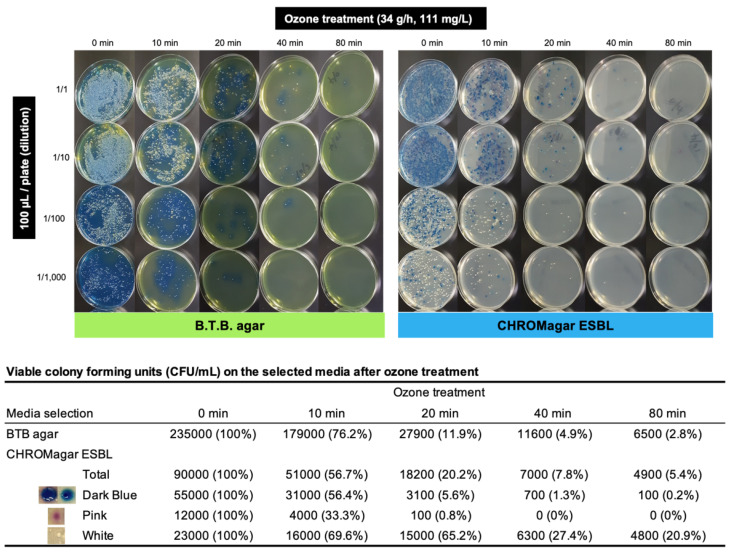
Isolation of bacteria from ozone-treated wastewater samples on B.T.B. agar and CHROMagar ESBL. An aliquot (100 µL) of ozone-treated wastewater sample was spread on the agar plate at the indicated dilution. Colony forming units (CFU/mL) were determined at the appropriate dilution for each time-point of ozone treatment. A colony on CHROMagar ESBL plate exhibited variable pigmentation, namely dark blue, pink, or white.

**Figure 3 antibiotics-11-00862-f003:**
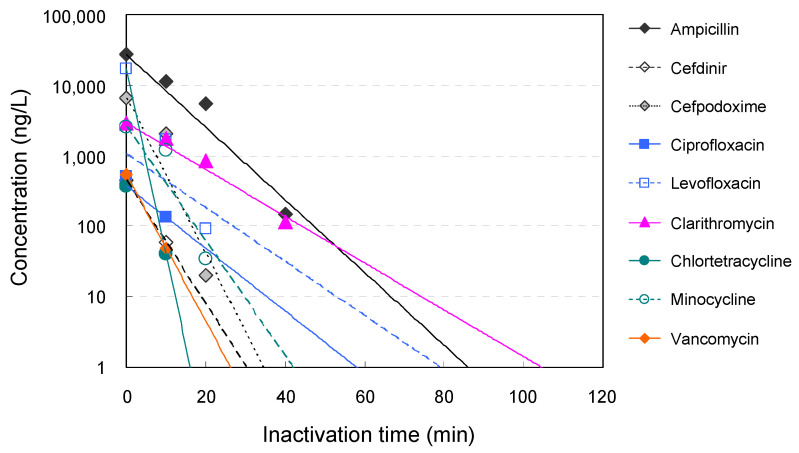
Time course of the concentration of antimicrobials in hospital wastewater during ozone treatment. Removal of antimicrobials over time during ozone treatment of hospital wastewater. The vertical axis shows the logarithmic value of the detected concentration, and the horizontal axis shows the time after the start of treatment.

**Table 1 antibiotics-11-00862-t001:** Metagenomic DNA-seq analysis of bacteria trapped on a 0.2 µm filter after ozone treatment.

Ozone Treatment (Min)		0	10	20	40	80
**DNA Concentration (ng/µL)**		0.5	1	<0.1	<0.1	<0.1
**Metagenome DNA-Seq (Total Reads)**		1,544,832	2,845,016	79,798	12,568	3060
**Megablast Search of Bacteria (Genus) ***						
	*Bacteroides*	141,342	366,044	19	42	18
	*Parabacteroides*	25,428	60,011	1	4	4
	*Acidovorax*	14,454	25,271	1	195	28
	*Aeromonas*	4125	18,455	2	41	6
	*Citrobacter*	3939	7889	0	18	2
	*Escherichia*	13,791	27,109	6	9	0
	*Klebsiella*	13,049	24,336	8	23	5
	*Raoultella*	12,630	14,533	0	37	12
	*Acinetobacter*	19,510	30,185	4	73	29
	*Pseudomonas*	10,162	14,073	0	528	213
	*Bifidobacterium*	21,541	41,180	8	5	0
	*Enterococcus*	3322	5245	0	0	2
	*Ruminococcus*	17,829	33,639	2	8	2

* Next-generation sequencing-read counts for the detected notable bacterial genera are shown.

**Table 2 antibiotics-11-00862-t002:** Concentrations of targeted antimicrobials in hospital wastewater during ozone treatment (N.D.: Not detected).

Classification	Antimicrobials	Treatment Time (Min)				
		0	10	20	40	80
*β*-lactams	Ampicillin	27,106	11,366	5522	148	N.D.
	Cefdinir	443	59	N.D.	N.D.	N.D.
	Cefpodoxime	6603	2040	20	N.D.	N.D.
	Cefpodoxime proxetil	N.D.	N.D.	N.D.	N.D.	N.D.
	Ceftiofur	N.D.	N.D.	N.D.	N.D.	N.D.
New quinolones	Ciprofloxacin	505	134	N.D.	N.D.	N.D.
	Levofloxacin	16,818	1676	92	N.D.	N.D.
Macrolides	Azithromycin	N.D.	N.D.	N.D.	N.D.	N.D.
	Clarithromycin	2933	1724	832	114	N.D.
Tetracyclines	Chlortetracycline	373	4	N.D.	N.D.	N.D.
	Doxycycline	N.D.	N.D.	N.D.	N.D.	N.D.
	Minocycline	2577	1185	35	N.D.	N.D.
	Oxytetracycline	N.D.	N.D.	N.D.	N.D.	N.D.
	Tetracycline	N.D.	N.D.	N.D.	N.D.	N.D.
Glycopeptides	Vancomycin	541	50	N.D.	N.D.	N.D.

## Data Availability

All raw read sequence files are available from the DRA/SRA database (accession numbers DRR376744–DRR376803 (see Appendix A)).

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
