# Peer review of "Inactivation of Bacteria and Residual Antimicrobials in Hospital Wastewater by Ozone Treatment"

_antibiotics, 2022, doi:10.3390/antibiotics11070862_

Round 1

Reviewer 1 Report

The topic of the article is very important for ensuring water quality. It deals with an issue that is serious and so far without sufficient solutions. The article is written at a very high professional level. It provides a wealth of relevant verifiable data for use in practice. I only have one comment, more technical. Why is the Method and Material section included at the end of the article?

Author Response

Comments and Suggestions for Authors

The topic of the article is very important for ensuring water quality. It deals with an issue that is serious and so far without sufficient solutions. The article is written at a very high professional level. It provides a wealth of relevant verifiable data for use in practice. I only have one comment, more technical. Why is the Method and Material section included at the end of the article?

Response:

Thank you for the comment and the evaluation. According to this comment and a relevant suggestion made by Reviewer 3, the Materials and Methods section has been moved and placed before the Results and Discussion sections.

Reviewer 2 Report

In the present study, the authors have described the effectual removal of different antimicrobials as well as certain bacteria. Though they determine ARGs as well despite giving results did not elaborate on them. To me discussing ARGs was more important than discussing bacteria in this regard. Additionally, I have a number of suggestions:

In the introduction part and even the abstract, it was mentioned that, To the best of our knowledge, this is the first report on the effectiveness of ozonation in the removal of antimicrobial-resistant bacteria and antimicrobials, I think you should better search on google scholar. This might be the first study in Japan, or the other people have studied different antimicrobials and bacteria, that can make it innovative not the use of ozone against these two things.

1.     Line 64: treatment of hospital wastewater…….. as hospital wastewater is already treated as you mentioned before so better to mention the treatment of AMR rather than just treatment

2.     Line 94: Other bacteria genera were also reduced to 0.02% of the original number???? Do you want to say reduced up to 0.02% of the original number?

3.     Table 1: some of the genera were zero after 20 minutes but the number was increased after 40 minutes of treatment, was it the same wastewater? Or do you treat one sample for 20 minutes and another for 40 minutes? How can the same sample goes from zero to a higher value again, do you have any explanation for that?

4.     Line 121: B.T.B stands for?

5.     I suggest you hire some native speakers to polish the language, there are many sentences that should be corrected and I am not mentioning all of them so kindly ask some native speakers to check for language mistakes.

6.     I was thinking why don’t you take the sample to extract DNA and check for the concentration of DNA or different species, why do you take the DNA from plates rather than doing a water sample analysis

7.     Studying ARGs is a very important part of such studies and you placed that data into supplementary materials, what is the reason for that? You can see all the good articles, they put results about ARGs as the part of the main body of the article

8.     Line 218: ARGs, when you put the word for the first time, write its full form.

9.     I am not satisfied with the discussion part, the discussion should be a comparison of your study with the previous, you did not compare any results to the previous studies as it should be, please add more references and comparisons to validate the significance of this study.

10.  The concentration of antibiotics was determined by UPLC-MS/MS, so it was based on mass or you have used some internal standards, and if it was MS does the metabolites detected? Why MS?

11.  Would you please write the Conclusion of this study under some separate heading, I could not find the conclusion?

Author Response

Comments and Suggestions for Authors

In the present study, the authors have described the effectual removal of different antimicrobials as well as certain bacteria. Though they determine ARGs as well despite giving results did not elaborate on them. To me discussing ARGs was more important than discussing bacteria in this regard. Additionally, I have a number of suggestions:

In the introduction part and even the abstract, it was mentioned that, To the best of our knowledge, this is the first report on the effectiveness of ozonation in the removal of antimicrobial-resistant bacteria and antimicrobials, I think you should better search on google scholar. This might be the first study in Japan, or the other people have studied different antimicrobials and bacteria, that can make it innovative not the use of ozone against these two things.

Response:

As per the comments made by the reviewer, we have carefully reviewed similar case studies. Indeed, we are aware of the papers published abroad on the topic of ozonation in the removal of antimicrobial-resistant bacteria and antimicrobials from hospital wastewater. However, there still have been no reports on the actual implementation of ozone treatment systems in hospital facilities and evaluation on a practical scale, even from a global perspective. We presume that this contributes to the novelty and the usefulness of this study. On the other hand, the treatment system investigated for implementation in this study is a batch-type treatment system that temporarily stores hospital wastewater in a separate tank, which poses certain challenges in terms of practical operation. Therefore, as a next step to this study, we are currently in the process of developing a continuous treatment system capable of treating large volumes of wastewater without interruption, which we believe will be highly effective from a practical standpoint. According to the suggestion and the abovementioned aspects, the description in the manuscript has been improved to make the novelty of this study clear.

  1. Line 64: treatment of hospital wastewater…….. as hospital wastewater is already treated as you mentioned before so better to mention the treatment of AMR rather than just treatment

Response:

As suggested by reviewer, the text has been revised to ‘treatment of AMR-related factors in hospital wastewater before discharging’.

  1. Line 94: Other bacteria genera were also reduced to 0.02% of the original number???? Do you want to say reduced up to 0.02% of the original number?

Response:

As suggested by reviewer, the text has been revised to ‘up to 0.02%’.

  1. Table 1: some of the genera were zero after 20 minutes but the number was increased after 40 minutes of treatment, was it the same wastewater? Or do you treat one sample for 20 minutes and another for 40 minutes? How can the same sample goes from zero to a higher value again, do you have any explanation for that?

Response:

We investigated the same wastewater sample at both time points. As shown in Figure 1, a 1 m3 volume of wastewater was transferred to the batch reactor, and ozone was continuously supplemented through a microbubble nozzle with circular pumping. We observed a few colonies after 40 to 80 min exposures as shown in Table 1, but during the circular pumping, such residual live bacteria may be peeled off the surface of the batch reactor tank. We believe that this is the reason for the limited number, which was negligible when compared with non-treated wastewater.

  1. Line 121: B.T.B stands for?

Response:

B.T.B stands for (Bromothymol Blue, lactose agar; Drigalski Agar, Modified). This description has been included in the text.

  1. I suggest you hire some native speakers to polish the language, there are many sentences that should be corrected and I am not mentioning all of them so kindly ask some native speakers to check for language mistakes.

Response:

As per the suggestion, language revision of the current version of the manuscript has been performed by native speakers, supported by Editage, with expertise in this area.

  1. I was thinking why don’t you take the sample to extract DNA and check for the concentration of DNA or different species, why do you take the DNA from plates rather than doing a water sample analysis

Response:

DNA concentration obtained from raw wastewater is shown in Table 1. It could be one of the indicators that help estimate the efficacy of ozone treatment. Besides, we assessed the bacteria CFU on a non-selective agar plate. Therefore, we suppose that these would be sufficient to reach our conclusion.

  1. Studying ARGs is a very important part of such studies and you placed that data into supplementary materials, what is the reason for that? You can see all the good articles, they put results about ARGs as the part of the main body of the article

Response:

We have included a supplementary Table (Table S3) showing the detected RPKM value for each OTU of ARG (AMROTU) after the ozone treatment.

  1. Line 218: ARGs, when you put the word for the first time, write its full form.

Response:

The full form of ARG has been included in the text.

  1. I am not satisfied with the discussion part, the discussion should be a comparison of your study with the previous, you did not compare any results to the previous studies as it should be, please add more references and comparisons to validate the significance of this study.

Response:

We have improved the discussion section by citing previous publications as below.

  • This finding is consistent with the previous reports (PMID: 35203813; PMID: 34972828; PMID: 27058129), indicating that it supports the proposal for installation of an ozone treatment system with the hospital wastewater tank.

  1. The concentration of antibiotics was determined by UPLC-MS/MS, so it was based on mass or you have used some internal standards, and if it was MS does the metabolites detected? Why MS?

Response:

As pointed out by the reviewer, the measurement of antimicrobial residues in wastewater was based on MS. This is because the concentration of antimicrobial residuals in wastewater is generally extremely low, of the order of ng/L, and cannot be detected by conventional methods such as UV absorbance determination. Furthermore, the MS/MS system makes it possible to detect trace concentrations of ng/L. Internal standards are often not available from reagent companies, especially for pharmaceuticals, and identification and recovery correction are performed with and without antimicrobial standard materials in accordance with previously reported data. In this context, we analyzed the antibiotics itself and not their metabolites. However, we believe that the reviewers’ perspective is important in the context of environmental risk management of pharmaceuticals in the environment. We intend to conduct further investigations on the occurrence and the environmental fate of antimicrobial metabolites in the water environment in near future. According to the suggestion, a description of the detailed methods used for the detection and quantification of antimicrobials in wastewater has been included in 2.6 Analytical procedures for antimicrobials.

  1. Would you please write the Conclusion of this study under some separate heading, I could not find the conclusion?

Response:

According to this suggestion, a description on the conclusions of the present study has been included in section ‘Conclusions’ in the revised manuscript as follows:

“A plant-scale ozone wastewater treatment facility based on a batch-type process was implemented in a core hospital located in the heart of Japan, and the inactivation effect on antimicrobial-resistant bacteria and antimicrobial residues was evaluated. The ozone treatment was effective in inactivating both antimicrobial-resistant bacteria and antimicrobials in hospital wastewater. Even when direct treatment, without filtration or biological pretreatment, was employed, most of the clinically problematic antimicro-bial-resistant bacteria and residual antimicrobials were inactivated within 20−40 min of direct ozone treatment. The fact that a variety of antimicrobial-resistant bacteria and antimicrobials were detected at high concentrations in the hospital wastewater is a key point related to the development and spread of AMR. The fact that these organisms and antimicrobials can be inactivated by advanced wastewater treatment is significant in terms of taking feasible and effective countermeasures for addressing AMR in the environment.

The overall results signify a novel approach for preventing the environmental risks associated with the spread of AMR and could facilitate the early detection of nosocomial infection risk and the reduction of the environmental impact of AMR. Our findings could help enhance the effectiveness of introducing advanced wastewater treatment systems, not only at wastewater treatment plants but also at medical facilities, to reduce the dis-charge of pollutants into rivers, thereby contributing to the safety of environment and human health.”

Reviewer 3 Report

Removal of drug-resistant microorganisms and pharmaceutical residues from the aquatic environment, including sewage, is a contemporary challenge for the world of science. There have been a number of scientific publications devoted to this issue in recent years. In this context, the article presented for review, on the destructive effect of ozone on some types of antimicrobial-resistant bacteria and residual antimicrobials contained in hospital wastewater falls within this issue. In the opinion of the Reviewer, the article is a valuable contribution to broadening the knowledge that can applied in practice. Nevertheless, before publishing the article, the Reviewer believes that the following issues should be clarifield and suplemented:

1/ For editorial reasons, the Reviewer proposes to include p.2 "Results" and next p.3 "Discussion" after p.4 "Material and Methods",

2/ In the content of p. 2.1, p.2.3 and in Tab. 1 and 2, Fig.3 the range of changes in direct ozonation time is given, but there is no information on the applied of ozone doses. The dose of ozone depends on the concentration of organic matter (measured, for example, with the DOC index) present in the wastewater, which reacts with it. For this reason, the ozone dose is linked to DOC in form of O3/DOC,

3/ Explain the fluctuations (especially increases) in number of bacteria (data from Tab.1) and concentration of antimicrobials (data from Tab.2) with prolonged ozonation times,

4/ The reaction of ozone with pharmaceuticals ofen produces byproducts of oxidation of these substances, which are also hazardous to the envinonment. Therefore, it is necessary to identify there in the wastewater after ozonation, as well as to determine their toxic effect on the ecosystem. This information is not available in the article.

Author Response

Comments and Suggestions for Authors

Removal of drug-resistant microorganisms and pharmaceutical residues from the aquatic environment, including sewage, is a contemporary challenge for the world of science. There have been a number of scientific publications devoted to this issue in recent years. In this context, the article presented for review, on the destructive effect of ozone on some types of antimicrobial-resistant bacteria and residual antimicrobials contained in hospital wastewater falls within this issue. In the opinion of the Reviewer, the article is a valuable contribution to broadening the knowledge that can applied in practice. Nevertheless, before publishing the article, the Reviewer believes that the following issues should be clarified and supplemented:

1/ For editorial reasons, the Reviewer proposes to include p.2 "Results" and next p.3 "Discussion" after p.4 "Material and Methods",

Response:

According to this suggestion and a similar suggestion made by Reviewer 1, Materials and Methods section has been placed before the Results and Discussion sections.

2/ In the content of p. 2.1, p.2.3 and in Tab. 1 and 2, Fig.3 the range of changes in direct ozonation time is given, but there is no information on the applied of ozone doses. The dose of ozone depends on the concentration of organic matter (measured, for example, with the DOC index) present in the wastewater, which reacts with it. For this reason, the ozone dose is linked to DOC in form of O3/DOC.

Response:

Prior to this investigation, dissolved ozone and gaseous ozone concentrations were monitored for reference. Dissolved ozone and flue gas ozone concentrations were below the detection limit at the time when the majority of antimicrobial resistant bacteria and antimicrobials are inactivated and begin to be detected after several hours, much longer than the treatment time described in the text. Since the measurement of organic matter in this study was on a COD basis, it cannot be evaluated on an O3/DOC basis. However, to facilitate understanding of the results of this study, a table of water quality parameters such as COD, BOD, SS, and Coliform group, which were measured in this study, has been added to the supplemental materials as Table S1, and additional notations have been included in the revised manuscript.

3/ Explain the fluctuations (especially increases) in number of bacteria (data from Tab.1) and concentration of antimicrobials (data from Tab.2) with prolonged ozonation times,

Response:

We investigated the same wastewater sample at both time points. As shown in Figure 1, a 1 m3 volume of wastewater was transferred to the batch reactor, and ozone was continuously supplemented through a microbubble nozzle with circular pumping. We observed a few colonies after 40 to 80 min exposures as shown in Table 1, but during the circular pumping, such residual live bacteria may be peeled off the surface of the batch reactor tank. We believe that this is the reason for the limited number, which was negligible when compared with non-treated wastewater.

4/ The reaction of ozone with pharmaceuticals often produces byproducts of oxidation of these substances, which are also hazardous to the environment. Therefore, it is necessary to identify there in the wastewater after ozonation, as well as to determine their toxic effect on the ecosystem. This information is not available in the article.

Response:

As the reviewer pointed out, previous studies have reported that ozone treatment causes the decomposition of environmental chemicals and produces byproducts. As for these byproducts, it has been shown that their lower molecular weight generally makes them less toxic than the parent compound, but there have been reports of increased toxicity for some substances. As far as the authors perceive, the antimicrobials examined in this study do not contain components whose toxicity increases by ozone treatment compared to that before the treatment. Even when harmful byproducts are generated, they presumably become harmless after a sufficient treatment time due to the strong oxidizing power of ozone. Due to technical limitations, bioassay or other ecotoxicity impact assessments were not conducted in this study, and we consider that this part of the study remains a challenge for future investigations. According to the suggestion, a discussion on byproducts has been added to the revised manuscript along with additional references.

Round 2

Reviewer 2 Report

I am happy the authors have well described and incorporated all the comments, so I suggest to accept the article in the present form now.

Reviewer 3 Report

The Authors of the paper improved the manuscript in accordance with Reviewer suggestions so the Reviewer believes that manuscript can be published in present form.